# Lipid-Driven Immunometabolism in Mesenchymal Stromal Cells: A New Axis for Musculoskeletal Regeneration

**DOI:** 10.3390/ijms262010117

**Published:** 2025-10-17

**Authors:** Vibha Velur, Patrick C. McCulloch, Francesca Taraballi, Federica Banche-Niclot

**Affiliations:** 1Center for Musculoskeletal Regeneration, Houston Methodist Academic Institute, Houston Methodist Research Institute, 6670 Bertner Ave., Houston, TX 77030, USA; 2Orthopedics and Sports Medicine, Houston Methodist Hospital, 6445 Main St., Houston, TX 77030, USA

**Keywords:** lipids, immunometabolism, mesenchymal stromal cells, tissue regeneration, post-traumatic osteoarthritis, osteoporosis, inflammation

## Abstract

The immunosuppressive and anti-inflammatory potential of mesenchymal stromal cells (MSCs) underpins their therapeutic value in musculoskeletal disorders. However, the underlying mechanisms remain ill-defined. Traditionally associated with immune cells, immunometabolism (the cellular metabolism–immune system interplay) is now recognized as central in a broader range of processes, including tissue homeostasis, repair, and chronic inflammation. Depending on the context and cell type, distinct metabolic pathways (e.g., fatty acid oxidation, lipid mediator biosynthesis) can drive pro-inflammatory/pro-resolving immune phenotypes. This dynamic is salient in musculoskeletal tissues: macrophage polarization, T-cell activation, and MSC immunomodulation are governed by metabolic cues. Emerging evidence highlights lipid-driven immunometabolism as a key player in MSC function, particularly in post-traumatic osteoarthritis (PTOA) and osteoporosis (OP). Unlike immune cells, MSCs rely on distinct metabolic programs (e.g., lipid sensing, uptake, and signaling) to exert context-dependent immunoregulation. In PTOA, persistent inflammation triggers lipid-centric metabolic pathways, enhancing MSC-driven immunomodulation and therapeutic outcomes. In OP, low-grade inflammation and altered lipid metabolism impair bone regeneration, modulating lipid-driven routes that can restore MSC osteogenic function and influence osteoclast precursors. This review explores how lipid-derived mediators and signaling contribute to MSCs’ immunosuppressive capacity, positioning lipid immunometabolism as a novel axis for rebalancing the inflamed joint microenvironment and encouraging musculoskeletal regeneration.

## 1. Introduction

Musculoskeletal disorders, including post-traumatic osteoarthritis (PTOA), rheumatoid arthritis, and osteoporosis (OP), affect millions of people and represent a significant and growing burden for healthcare systems worldwide. These disorders are not only a major cause of pain and disability but also present complex clinical challenges due to their chronic inflammatory nature and the limited regenerative capacity of joint tissues. Despite advances in surgery, pharmacological management, biomaterial, and cell-based therapies, the ability to restore damaged tissues remains far from ideal and current therapies often fail to address the root causes of tissue degeneration. This is especially significant in chronic or inflammatory contexts where the immune system plays a central, and often overlooked, role. For years, inflammation was seen mostly as a problem to suppress. But a growing body of evidence is reshaping this narrative: when properly regulated, immune responses can actually support tissue regeneration. In this landscape, regenerative strategies that can actively modulate the immune environment while promoting tissue repair are urgently needed.

This shift in perspective has given rise to a fascinating field at the crossroads of immunology and metabolism: immunometabolism. Once considered a niche concept, it has rapidly emerged as a key framework to understand how the metabolic state of immune and stromal cells influences their functions and fate. Crucially, this interaction is bidirectional: immune activation reshapes cellular metabolism, and metabolic cues, in turn, reprogram immune responses. Although immunometabolism is rapidly gaining ground in areas like cancer [1], cardiovascular [2], and autoimmune diseases [3], its application to musculoskeletal regeneration remains surprisingly limited.

Among the various metabolic players of this intricate network, lipids have gained increasing attention. Traditionally regarded as passive energy stores or structural membrane components, lipids are now recognized as potent signaling molecules that finely tune both inflammation and healing processes. Lipid mediators such as prostaglandins, specialized pro-resolving mediators (SPMs), and bioactive fatty acids (FA) like omega-3 derivatives are now known to fine-tune macrophage polarization, can influence mesenchymal stromal cell (MSC) immunomodulatory functions, and orchestrate the delicate balance between tissue degradation and regeneration [4,5,6,7]. In this review, we adopt the term “stromal” rather than “stem” for MSC, consistent with the International Society for Cell & Gene Therapy (ISCT) recommendations, to emphasize their broader, heterogeneous nature and predominant supportive and immunomodulatory roles rather than their stemness potential.

Despite these insights, the lipid-driven mechanisms that underlie musculoskeletal healing along with all their complexity and therapeutic promise remain largely underexplored in the current literature. Most regenerative strategies in orthopedics have traditionally focused on structural repair (e.g., scaffolds, cells, or growth factors) often overlooking the immunometabolic context in which healing occurs. This gap is particularly striking when considering that MSCs, a cornerstone of many orthopedic regenerative approaches [8,9,10,11,12,13,14,15,16], are themselves metabolically active cells whose immunoregulatory properties are deeply influenced by lipid signaling. Similarly, the inflamed joint environment is rich in lipid mediators that can either sustain damage or foster resolution, depending on their nature and timing.

This review aims to bring lipid-driven mechanisms into the spotlight, highlighting their emerging role in shaping the immune–regenerative interface within the musculoskeletal system. By dissecting how specific lipid species and metabolic pathways influence immune responses and MSC behavior, potentially acting as levers for promoting regeneration in damaged joints and tissues, we aim to uncover new opportunities for targeted therapeutic interventions that go beyond symptom management and structural repair. Rather than focusing solely on cells or scaffolds, we advocate for a more integrated perspective that also includes the biochemical “language” cells use to communicate to each other, resolve inflammation, and orchestrate regeneration. Understanding these mechanisms is not only timely; it is essential to designing next-generation treatments capable of regenerating and restoring tissue function while actively resolving chronic inflammation at its source.

## 2. Metabolic Demands in Skeletal Tissues

Musculoskeletal tissues, though structurally distinct, share a reliance on precise metabolic programming to maintain homeostasis, adapt to mechanical loading, and respond to injury. The metabolic needs of bone and cartilage are not static; they shift dynamically based on differentiation stage, inflammatory tone, oxygen availability, and biomechanical cues. Importantly, these tissues are increasingly recognized as metabolically active, integrating bioenergetic cues into transcriptional, epigenetic, and immunological decisions that ultimately govern regeneration or degeneration.

Bone is a highly vascularized and continuously remodeled by metabolically distinct osteoblasts, osteoclasts, and osteocytes. Osteoblasts, the matrix-producing cells derived from mesenchymal progenitors, transition from ATP-intensive aerobic glycolysis that supports collagen synthesis to oxidative phosphorylation (OXPHOS) with increased mitochondrial biogenesis during maturation, sustaining ECM mineralization [17,18]. This shift is coordinated by Wnt/β-catenin signaling viamTORC1, ATF4, and PGC-1α [19,20,21]; disrupted plasticity impairs osteoblast function and contributes to age-related loss and glucocorticoid-induced OP [22]. Osteoclasts, derived from monocyte-macrophage lineages, are responsible for bone resorption. Their metabolic phenotype is primarily glycolytic due to fuel proton pumping and lysosomal enzyme secretion [23], with fatty acid oxidation (FAO) and glutaminolysis supporting maturation and sustained resorptive activity [20,24,25]. Osteocytes, the most abundant bone cell type embedded within the mineralized matrix, act as mechanosensors that regulate both osteoblast and osteoclast function. They exhibit lower basal metabolism but are highly responsive to redox state, nutrient deprivation, and mitochondrial stress. Under hypoxia, osteocytes activate AMPK and stabilize HIF-1α to promote autophagy and preserving cellular integrity [26]. Emerging evidence suggests that osteocytic mitochondrial dynamics dysregulate bone turnover and may be impaired in metabolic bone diseases [27,28].

Unlike bone, articular cartilage is avascular, aneural, and resides in a physiologically hypoxic (~1–5% O_2_) niche. Chondrocytes rely mainly on anaerobic glycolysis for proteoglycan and type II collagen synthesis and downregulate OXPHOS to limit oxidative stress; HIF-1α dives GLUT1 and lactate dehydrogenase A (LDHA) this program [29,30]. In inflammatory states such as osteoarthritis (OA) or PTOA, pro-inflammatory cytokines such as IL-1β, TNF-α, and IFN-γ, reprogram metabolism, induce mitochondrial dysfunction, suppress SOX9-mediated chondrogenesis, and upregulate MMP-13 and ADAMTS-5, driving matrix degradation [31,32,33]. Notably, HIF-2α (a paralog of HIF-1α) promotes catabolic gene expression, linking inflammatory stress to metabolic remodeling in diseased cartilage [34]. Recent metabolomic and spatial transcriptomic studies reveal zonal heterogeneity within cartilage: a glycolysis- and antioxidant- enriched superficial zone resilient to shear stress, versus a deeper zone with greater mitochondrial respiration but higher vulnerability to hypoxia-induced apoptosis [35]. These functional layers underscore the need for regionally adaptive therapies that account for spatial metabolic heterogeneity in cartilage regeneration.

The osteochondral interface integrates biomechanical and metabolic signals across steep gradients of oxygen tension, pH, mechanical load, and nutrient diffusion, and is an early site of degeneration in OA and PTOA [36]. Subchondral bone remodeling—common in early OA—alters the vascular architecture, increasing angiogenesis and permeability, which perturbs the avascular cartilage environment. Mediators such as VEGF, RANKL, and IL-6 released from osteoblasts perturb the cartilage zone, drive inflammation, disrupt metabolic homeostasis, and accelerate ECM degradation [37]. Concurrent mechanical overload and matrix fragmentation further destabilize mitochondrial dysfunction and reactive oxygen species (ROS) accumulation.

Successful repair hinges on re-establish metabolic reciprocity between bone and cartilage. Regeneration of these units therefore necessitates dual-compartment solutions: osteogenic cues and mitochondrial activation for bone, and glycolytic support with anti-catabolic modulation for cartilage. Biomimetic scaffolds increasingly aim to recreate gradients of oxygenation, stiffness, and metabolic cues at the osteochondral interface, with zonal delivery of lipid mediators, oxygen carriers, or MSC-derived extracellular vesicles (EVs) with metabolic cargo offers promising for restoring homeostasis [38,39].

## 3. MSC Plasticity in Musculoskeletal Disease: Cellular Reprogramming Underlying Immunomodulatory Functions

MSCs are key players in musculoskeletal tissue regeneration due to their adaptive immunomodulatory property [12]. They sense inflammatory and damage-associated cues across joint, tendon, muscle, and marrow niches and rapidly adopt regulatory states that contain injury, promote resolution, and support structural repair [40] (Figure 1). This adaptability, more than lineage markers per se, drives in vivo performance across musculoskeletal conditions. In OA and PTOA, synovial fluid and peri-articular tissues exhibit persistent low-grade inflammation with hypoxic and oxidative stress that impedes resolution [41]; rheumatoid arthritis and chronic tendinopathies instead exhibit higher inflammatory tone with maladaptive stromal–immune loops [42]. By contrast, OP features chronic, low-grade inflammation characterized by estrogen deficiency and glucocorticoids, skewing the marrow niche toward osteoclastogenesis [43]. In all cases, the real-world efficacy of MSCs in musculoskeletal disorders hinges on how they sense local signals and reprogram their metabolism and secretome into a pro-resolving configuration via pattern-recognition receptors.

In particular, MSCs read immune-stromal signals through a layered receptor network: cytokines (e.g., IFN-γ, TNF-α, IL-1α/β, IL-6 family signals via gp130, and IL-17) via cognate receptors; chemokine (CCL2, CCL5, CXCL8/IL-8, CXCL12) via CCR/CXCR gradients [44,45]. These triggers are the most often exploited in preclinical and translational workflows. Moreover, complement anaphylatoxins (C3a/C5a) and extracellular nucleotides (P2X/P2Y) engage JAK–STAT, NF-κB, and MAPK nodes while lipid mediators (e.g., leukotrienes like LTB_4_, select prostanoids) and matrix-derived matrikines (biglycan, decorin) report tissue injury via TLRs and integrins [46,47,48,49]. Inflammation also imposes metabolic–biophysical constraints of the lesion microenvironment as local hypoxia, redox shifts, extracellular acidosis via microvascular perfusion defects (e.g., heightened oxygen consumption, oxidative bursts, and glycolytic lactate/H^+^ export compounded by CO_2_ hydration and impaired proton clearance), which can precede and trigger sterile inflammation, forming a feed-forward loop [50].

Under hypoxia, MSCs adopt a glycolysis-high, Warburg-like state driven by HIF-1α and AMPK: increased glucose uptake and flux (GLUT1, HK2, PFKFB3, LDHA), PDK1-mediated phosphorylates inhibition, reduced acetyl-CoA entry into the TCA cycle, pyruvate-to-lactate conversion with MCT4 export, constrained OXPHOS, and accompanying shift to a more reduced redox state (lower NAD^+^/NADH) buffered by LDHA-dependent NAD^+^ regeneration [51,52]. In parallel, DRP1-dependent fission and mitophagy (PINK1/Parkin; BNIP3/NIX) clear damaged mitochondria while reduced SIRT1/3 activity blunts PGC-1α programs; together reinforcing a glycolytic, stress-tolerant, immunomodulatory configuration [53,54].

On the other hand, extracellular acidosis (lactate and CO_2_ via carbonic anhydrase) allosterically modulates key enzymes, increases monocarboxylate transport capacity (e.g., MCT4), and engages dedicated pH sensors (e.g., acid-sensing ion channels and proton-sensing G-protein–coupled receptors, GPRs) [55]. Low pH further attenuates mitochondrial respiration, biases ATP production to glycolytic, and promotes lactate/H^+^ and ATP release via MCTs and pannexin/connexin channels, favoring adenosine-centered signaling (CD39/CD73 to A2A/A2B) that lower inflammatory tone [56,57,58].

The tissues’ microdamage also leaves a fingerprint in MSCs stress-response programs due to the release of pathogen- and damage-associated patterns (PAMPs and DAMPs) in the milieu in combination with a SASP from osteocytes and altered mechanotransduction under reduced load [59]. MSCs state is tuned by the engagement of TLR2/3/4/7/8/9 by PAMPs while TLRs and RAGE are involved through DAMPs and alarmins such as HMGB1, S100A8/A9, HSPs, hyaluronan or fibronectin-EDA fragments signal.

Meanwhile, the extracellular matrix speaks via stiffness, load, topography, and three-dimensionality architecture; direct contacts with synoviocytes, chondrocytes, fibroblasts, and immune cells rapidly reposition MSCs effector programs [60].

Once the sensing phase locks in, MSCs shift contact-depending signals and the paracrine secretome toward a tolerogenic profile deploying mediators and checkpoints including IDO/kynurenine, the adenosinergic axis (CD39/CD73), PGE_2_ (COX-2), IL-10, TGF-β, TSG-6, and immune-checkpoint molecules (PD-L1, HLA-G, galectin-9) that raise activation thresholds in inflamed tissues [61,62]. These outputs dampen innate pro-inflammatory circuits, promote M2-like macrophages, and limit dendritic cell maturation [61,62]. On the adaptive site, MSCs curb Th1/Th17 responses, reduce neutrophil activation and NETosis (i.e., a process where neutrophils release a web-like structure made of DNA and protein to immobilize and suppress microbes but can also escalate inflammation and collateral tissue damages [63]) and support Treg induction [61,62].

In bone disease, MSCs help rebalancing the RANKL/OPG rheostat against osteoclastogenesis via indirect and direct routes. Immunoregulation lowers RANKL-inducing cytokines and restores osteomacs (i.e., specialized bone-resident macrophages that line bone surfaces, support osteoblast, and coordinate remodeling [64]), while stromal crosstalk increases OPG, restraining osteoclastogenesis and preserving osteoblast function [65,66]. By attenuating drivers of Wnt antagonists (DKK1, SOST, SFRPs), MSCs facilitate Wnt/β-catenin activity and OPG transcription in osteoblast [67]. They may also enhance Wnt responsiveness (e.g., WNT3A/WNT10B) or R-spondins, cooperating with BMP2/BMP7–SMAD axes to amplify osteogenic programs and, indirectly, OPG expression [68].

Actionable strategies to harness MSC plasticity and actively reprogram their immunomodulatory ability are underway [69] and include pre-implantation licensing with IFN-γ, TNF-α or IL-1β (dose-titrated to preserve osteogenic competence), controlled hypoxia to enhance paracrine robustness [70,71], aggregate cells in 3D spheroids to boost potency and stress tolerance [72,73,74], and immuno-instructive biomaterials (e.g., collagen-based hydrogels and osteoconductive composites) that focus cues at the inflamed or imbalanced remodeling sites [13,14,60,75,76,77,78,79,80,81,82,83].

Beyond inputs and outputs, the missing link is the mechanism by which MSCs govern these programs. This resides in the so-called “immunometabolism”, with the lipid axis as a principal circuity that tunes MSC immunomodulation.

## 4. Lipid Immunometabolism of MSC in Musculoskeletal Health

Immunometabolism refers to the dynamic interplay between cellular metabolic pathways and immune function. In the context of MSC, immunometabolism signifies how shifts in energy utilization (e.g., glycolysis vs. FAO) and metabolic by-products influence the cells’ immunoregulatory behavior [84]. In this scenario, lipid metabolism has emerged to being tightly intertwined with MSC immunomodulatory potential, particularly in musculoskeletal tissues, and has evolved to encompass the way lipid-derived molecules govern this.

Lipid-driven immunometabolism specifically highlights the role of lipids as signaling modulators of cell behavior, inflammation, regeneration and tissue homeostasis [85]. Actually, lipids are no longer viewed merely as structural components or energy reserves. As broadly renowned, lipid metabolism, including both FA synthesis and oxidation, is central to regulating cellular behavior. FA synthesis, driven by key enzymes such as synthase and acetyl-CoA carboxylase, contributes to the production of membrane glycerophospholipids, which are essential for cell division and growth [86]. Concurrently, FAs are stored as triglycerides in lipid droplets and can be mobilized for mitochondrial oxidation to generate ATP via the Krebs cycle. Beyond their roles in proliferation and differentiation, both FA synthesis and oxidation profoundly influence immune cell function [87,88], highlighting the broader immunomodulatory potential of lipid metabolism. In this context, lipids like FAs and their derivatives are now recognized as potent and dynamic bioactive mediators of intra- and intercellular signaling, capable of activating specific intracellular and membrane-bound receptors that influence gene expression, cytokine production, and intercellular communication, thereby affecting how MSCs and immune cells communicate during inflammation and in turn regeneration [89]. The immunometabolic interface is especially critical in regenerative medicine, where inflammation must be tightly controlled to permit healing rather than fibrosis or chronic degeneration [89].

Interestingly, recent studies highlight PGE2 as one of the most extensively studied lipid-derived immunoregulatory mediators in MSCs, exerting immunosuppressive and anti-inflammatory effects through binding to prostaglandin receptors. It is synthesized from the long-chain polyunsaturated FA arachidonic acid (AA) via COX enzymes [90]. PGE2 is a key immunomodulatory factor secreted by MSCs in response to various immune cell types including T cells, NK cells, dendritic cells, and macrophages, and its secretion increases upon incubation with the TNF-α and IFN-γ [12]. To investigate the role of the lipidome in MSCs’ anti-inflammatory functions, Campos et al. conducted lipidomic profiling under pro-inflammatory conditions, observing a significant shift in phospholipid composition [89]. These findings indicate that distinct lipid species may actively contribute to MSC immunomodulatory functions, emphasizing the potential relevance of bioactive lipids in MSC biology [91,92].

Thus, lipid-driven immunometabolic pathways are emerging as crucial levers by which MSCs either promote a healing, anti-inflammatory milieu or, if dysregulated, contribute to pathology. In chronically inflamed or nutrient-altered states (such as obesity or aging), MSC immunometabolism may become “locked” in a dysfunctional mode, impairing their regenerative capacity. Conversely, harnessing immunometabolic plasticity (e.g., by supplying pro-resolving lipids) offers a strategy to dampen the pro-inflammatory setting of MSC and foster a pro-regenerative environment [93,94,95]. Further studies have identified that specialized lipid mediators, including resolvins (Rv), protectins, maresins (MaR) (derived from omega-3 FAs), and various oxylipins, orchestrate the resolution phase of inflammation, shifting the immune environment from a pro-inflammatory to a pro-regenerative state. These lipid mediators are biosynthesized through enzymatic pathways (e.g., lipoxygenases, cytochrome P450s), and their activity is highly context-dependent, governed by local availability of precursors (like eicosapentaenoic acid, EPA, and docosahexaenoic acid, DHA), hypoxic stress, and metabolic cues from the microenvironment [96,97,98].

In summary, lipid immunometabolism is the nexus of metabolism, immunity, and regeneration, serving as a fundamental component of MSC biology. It governs their ability to modulate the immune response, endure hostile environments, and orchestrate the musculoskeletal tissues repair process [91]. Advancing understanding of this metabolic axis and learning how to manipulate can offer a promising avenue for enhancing the efficacy of MSC-based therapies in regenerative medicine.

### 4.1. MSC Secretome, Lipidome and Bioactive Lipid Mediators in Regeneration

MSCs heal tissues largely through their secretome, the cocktail of soluble factors and vesicles they release includes not only proteins (cytokines, growth factors) but also bioactive lipids and lipid-containing exosomes [85]. The MSC lipidome—which is the full repertoire of lipid species available and released by MSCs—has gained attention for its role in modulating inflammation and immunity by specific receptors activations and metabolic cascade [91,92]. The ability of MSCs to sense and respond to lipid signals allows them to adapt to diverse tissue contexts, making them especially valuable in treating musculoskeletal conditions where chronic inflammation impedes healing as raised in PTOA. Over the past few years, there has been a marked shift from merely suppressing inflammation to actively promoting its resolution using lipid mediators. In this context, endogenous lipids derived from omega-3 and omega-6 FA have drawn major interest compared to traditional anti-inflammatories since they are not only able to dampen pro-inflammatory signals but also stimulate tissue repair and return to homeostasis.

Notably, the SPMs—including Rv (from EPA and DHA), MaR (from DHA), protectins and lipoxins—can profoundly influence MSC function and the tissue microenvironment [91]. For example, a recent study of AlZahari et al. examined Resolvin E1 (RvE1) and Maresin-1 (MaR1) in an inflammatory bone-loss model and found that these lipids enhanced MSC-mediated regeneration. Human bone marrow MSCs exposed to a pro-inflammatory stimulus normally show blunted osteogenic activity, but the addition of RvE1 or MaR1 counteracted this: treated MSCs had substantially increased mineralized bone formation, with the combination of RvE1 plus MaR1 producing the greatest osteogenesis under inflammatory conditions [99]. Mechanistically, RvE1 and MaR1 helped resolve the inflammatory milieu while directly boosting the osteogenic differentiation pathways in MSCs. RvE1, for instance, is known to inhibit osteoclastogenesis (bone breakdown) by modulating NF-κB signaling and preserving osteoprotegerin (OPG) levels, thereby tipping the balance toward bone formation. On the other hand, MaR1 has demonstrated the ability to reduce pro-inflammatory cytokines and restore MSC function in damaged tissues, supporting organ regeneration and healing. In the same study, MSCs treated with RvE1/MaR1 exhibited a shift toward an anti-inflammatory secretory profile: TNF-α and IFN-γ levels dropped, while anti-inflammatory cytokines like IL-10, TGF-β, and IL-4 rose, especially with combined RvE1 plus MaR1 treatment. This illustrates how ω-3-derived lipids can augment the immunosuppressive potency of MSCs, transforming a hostile inflammatory setting into one conducive for regeneration. Consistently, other SPMs such as Resolvin D1 (RvD1) have shown regenerative benefits in musculoskeletal models: repeated RvD1 administration in a bone allograft model significantly improved new bone integration, in part by directly enhancing osteoblast differentiation and reducing osteoclast activity, thus promoting healing of the graft [100]. Furthermore, SPMs have shown broad safety and the ability to protect joint tissues after injury in preclinical models, encouraging their translation into orthopedic therapies as “resolution pharmacology”. For example, intra-articular administration of MaR1 in mice after ACL rupture curtailed leukocyte influx, lowered inflammatory cytokines, and prevented PTOA development, improving cartilage histology and bone morphology scores [101]. Recent human studies also underscore this potential as well. In a 2023 pilot trial named GAUDI, knee OA patients who received SPM-enriched daily oil supplements experienced significant pain reduction and improved quality of life over 12 weeks, without adverse events [102]. Such evidence positions SPM pathways as a promising therapeutic avenue to avert chronic inflammation and degeneration following joint trauma.

Similarly, key bioactive lipid mediators derived from omega-6 polyunsaturated FAs (PUFAs) act as local hormones that can orchestrate the active resolution phase of inflammation [98]. For instance, AA, a membrane phospholipid-derived omega-6 PUFA, is converted by COX enzymes in MSCs into PGE_2_, a well-known immunosuppressive lipid [103]. PGE_2_ is recognized as a key factor secreted by MSCs upon interaction with immune cells (T cells, macrophages, dendritic cells, etc.), and it potently suppresses inflammatory responses by binding PGE_2_ receptors on those immune cells [84]. Through PGE_2_, MSCs can inhibit T-cell proliferation and skew macrophages toward an anti-inflammatory phenotype, thereby protecting tissues from excessive inflammatory damage. Another AA-derived mediator, lipoxin A4 (LXA_4_), has been implicated in MSC-driven regeneration. In particular, MSC secretion of LXA_4_ in an acute lung injury model reduced TNF-α levels and improved survival, an effect that could be reversed by blocking the LXA_4_ receptor [104,105].

Beyond AA products, MSCs can also be primed with omega-3 FA to improve their therapeutic secretome. In a 2024 study, adipose-derived MSCs were priming for 72 h with an omega-3 PUFA, DHA, and the optimal dose not only was well-tolerated, but significantly increased the cells biosynthetic and metabolic activity, yielding a secretome richer in proteins and growth factors [106]. The DHA-primed MSCs showed enhanced viability and even a bias toward adipogenic differentiation, but importantly, their conditioned medium more potently supported neuronal growth in vitro. While that study focused on neuro-regeneration, it underscores a broader point: lipid molecules like DHA can reprogram MSCs to secrete higher levels of therapeutic factors. Along similar lines, exosomes from MSCs cultured under metabolic stress (e.g., low oxygen to stabilize HIF-1α) have been shown to attenuate “inflammaging” in joint tissues and slow PTOA progression, highlighting the modulatory power of the MSC secretome’s lipid and protein cargo in an inflammatory joint environment [107]. Together, these findings emphasize that the MSC secretome/lipidome—including PGE_2_, lipoxins, Rv, MaR, and other oxylipins—plays a central role in tissue regeneration. By damping chronic inflammation and fostering resolution, these bioactive lipids enhance MSCs’ immunosuppressive capacity and create conditions for effective repair of cartilage, bone, and other musculoskeletal tissues. Harnessing such lipid mediators (either by delivering them directly or by inducing MSCs to produce them) is therefore a promising strategy to improve regenerative therapies for OA, OP, and even metabolic disorders.

Beyond OA, resolution biology is being explored in other musculoskeletal and metabolic disorders. Chronic, low-grade inflammation in OP has been attributed to a failure of resolution, and SPMs have shown efficacy in preclinical models of bone loss [108]. It is now hypothesized that boosting pro-resolving mediators could complement standard OP treatments by extinguishing inflammation that drives bone catabolism. Likewise, in metabolic conditions like obesity and diabetes (which often exacerbate OA and impair healing), omega-3 PUFA derivatives and SPMs have demonstrated anti-inflammatory and insulin-sensitizing effects. For instance, supplementing obese mice with 17-HDHA—a precursor to protectin D1—alleviated adipose inflammation and improved insulin sensitivity [109]; highlighting the systemic benefits of resolving mediators.

In summary, resolution biology has emerged as a key trend. By harnessing SPMs and their PUFA precursors, researchers are actively promoting the natural resolution of inflammation to reduce pain, protect cartilage and bone, and improve healing outcomes in musculoskeletal conditions. This paradigm shift—treating “inflammaging” and injury not just by blocking damage, but by restoring balance—is opening novel therapeutic directions for OA, OP, and beyond.

### 4.2. Key Lipid-Sensing Receptors in MSCs

MSCs express several lipid-sensing nuclear receptors that link external lipid signals to profoundly influence cell function and fate. The three key ones are PPARγ, GPR120, and TLR4, which respectively illustrate an anti-inflammatory metabolic regulator, an omega-3 FA membrane sensor, and an immune pattern-recognition receptor. These receptors help determine MSC fate and immunomodulatory profile in the context of musculoskeletal health.

#### 4.2.1. Peroxisome Proliferator-Activated Receptor Gamma (PPARγ)

PPARγ is the master transcriptional regulator of adipogenesis and FA metabolism [110]. It is ubiquitously expressed in MSCs and, despite the fact that it is known to tip toward adipogenesis at the expense of osteogenesis [111], recent studies confirm that inhibiting PPAR-γ can reverse this trend. However, the direction of its net effect depends on context (cell source, inflammatory tone, and dosing/time of modulation). Indeed, the PPARγ-driven adipogenesis is one reason bone marrow MSC in aged or glucocorticoid-treated animal models of OP, when treated with a PPAR-γ modulator such as BADGE, show significantly reduced bone marrow adiposity while enhancing osteoblast activity and bone formation [112]. However, such inhibition may also diminish PPARγ’s anti-inflammatory influence, so benefits on skeletal repair must be weighed against potential losses in immune regulation. Treated OP mice showed increased bone volume and elevated osteogenic markers alongside reduced adipogenic markers, indicating that targeting PPAR-γ skews MSC differentiation back toward the osteogenic lineage. These findings have also translational resonance, as PPAR-γ agonist drugs (TZDs for type-2 diabetes) often cause bone loss. Interestingly, this adverse skeletal impact contrasts with their systemic benefit in dampening inflammation, highlighting the receptor’s context-dependent actions. Magadum et al. found that blocking PPAR signaling in periodontal MSCs enhanced their bone differentiation capacity in an inflammatory environment, primarily through activation of the Wnt signaling pathway [113]. Additionally, emerging evidence suggests that PPAR also regulates glycolysis in MSC, thereby influencing their immunosuppressive functions and therapeutic efficacy [114,115]. Beyond lineage fate, PPARγ also carries broadly anti-inflammatory effects as it antagonizes NF-κB and other pathways in immune cells [116]. In MSCs, PPARγ’s anti-inflammatory role is less direct but evident in certain contexts. For instance, PPARγ activation in autoimmune disease leads MSC to increased expression of key enzymes and lipid transporters such as lipoprotein lipase, CD36, or enzymes involved in FAO [117,118]. These molecules play a crucial role in lipid metabolism and help maintain lipid homeostasis, particularly under conditions of metabolic or inflammatory stress. Moreover, through its regulatory control over lipid metabolism, PPARγ activation actively dampens the accumulation of pro-inflammatory lipid mediators such as oxidized FAs, prostaglandins, and leukotrienes. By downregulating their synthesis and enhancing their degradation or clearance, PPARγ helps shift the cellular environment toward an anti-inflammatory state, supporting immune resolution and tissue homeostasis [117,119]. Thus, two opposing levers coexist: inhibiting PPARγ can improve osteogenic regeneration yet forfeit some anti-inflammatory tone, whereas activating PPARγ can promote immune quiescence yet jeopardize osteochondral anabolism if overdone [119]. This protective metabolic role can, therefore, compete with the needs for tissue repair in skeletal disorders, implying that complete agonism may not be universally beneficial.

Meanwhile, attention has turned to PPAR-β/δ, another lipid-sensitive transcription factor expressed in MSCs. Intriguingly, PPAR-β/δ appears to regulate the chondroprotective and immunomodulatory functions of MSCs [120]. Loss-of-function studies illustrate the PPAR isotypes balance: knocking out PPARδ forces MSCs to rely more on glycolysis, which unexpectedly enhanced their immunosuppressive functions by increasing T-cell inhibition and anti-inflammatory factor production [121]. Yet, this metabolic shift does not necessarily guarantee improved matrix regeneration, underscoring trade-offs between immune modulation and structural repair. In an arthritis model, PPARβ/δ-deficient MSCs showed superior therapeutic effects due to this metabolic shift [122]. Intriguingly, a recent approach investigated the PPAR-β/δ activity in donor MSCs to predict their efficacy in reducing inflammation and supporting cartilage in OA [123]. In a PTOA mouse model, PPAR-β/δ was found to promote cartilage breakdown under injury conditions, yet if MSCs were treated with a PPAR-β/δ agonist, their chondrogenesis improved due to the receptor’s anti-inflammatory effects in the joint. Notably, knocking down PPAR-β/δ in MSCs enhanced their therapeutic potency in arthritis, partly by boosting the MSCs’ secretion of factors that drive macrophages from a pro-inflammatory (M1) to anti-inflammatory (M2) phenotype [124].

Overall, opposing outcomes can be mentioned within the PPAR axis: PPARγ’s immunomodulatory benefits can conflict with its pro-adipogenic/anti-osteogenic liabilities, while PPAR-β/δ’s support of oxidative metabolism and chondrogenesis can conflict with contexts where glycolytic reprogramming fosters stronger immunosuppression [125,126,127]. While the pivotal roles of PPARs in the immune system have been extensively reviewed elsewhere, direct evidence linking PPAR-mediated regulation of MSC immunometabolism remains limited, underscoring the need for further investigation. Indeed, nowadays, there is no single “best” direction of PPAR modulation and the optimal polarity likely shifts across disease stages (acute injury vs. chronic OA), target tissues (bone vs. cartilage), and therapeutic goals (immediate inflammation control vs. matrix rebuilding). This suggests that—rather than broad receptor activation or inhibition—selective pathway targeting along with fine-tuned, phase-specific, and possibly isoform-selective PPAR modulation may prolong and amplify the benefits of MSC therapy in OA and PTOA. Addressing the contradictions above will require comparative studies that stratify by MSCs source, inflammatory milieu, and timing/dose, and that co-monitor TLR4/NF-κB activity to map when PPAR activation vs. inhibition yields the best trade-offs for synovial inflammation control and osteochondral regeneration.

#### 4.2.2. GPR120-Free FA Receptor 4

Complementing the PPAR role in bone and cartilage biology is the emerging function of GPR120 (Free FA Receptor 4), a GPRs for unsaturated long-chain omega-3 FAs, especially DHA and EPA, acting as a mediator on metabolism [128]. In MSC biology, GPR120 serves as a molecular sensor and its activation biases MSC function and differentiation towards bone formation while restraining adipogenesis [129]. Importantly, GPR120 activity was shown to tilt the balance in favor of anabolic process by leveraging the body’s own lipid-sensing mechanisms, improving bone density and cartilage integrity. However, the anabolic effect of GPR120 activation appears highly dose- and context-dependent, and overstimulation could desensitize the receptor or disrupt the balance between bone and fat formation [129]. In osteoporotic OVX mice, treatment with the selective GPR120 agonist TUG-891 restored bone mass, microarchitecture, and bone formation rates [130]. These anabolic benefits are not uniform across conditions: in OA models (Anterior Cruciate Ligament Transection-induced OA in mice), loss of GPR120 exacerbated cartilage degeneration and subchondral bone changes, while DHA-mediated GPR120 activation reduced inflammatory responses in chondrocytes [131]. These opposing outcomes—loss worsening degeneration while activation favors protection—underscore that GPR120’s role is perhaps more homeostatic than purely anabolic. Its activation may restore balance under stress rather than simply enhance regeneration in all contexts. Upon DHA binding, GPR120 undergoes a conformational change that activate intracellular G proteins or β-arrestin2, initiating downstream signaling through PI3K/AKT and phospholipase C while inhibiting NF-κB pathways, which contribute to pro-regenerative and anti-inflammatory responses, respectively [132,133,134]. Notably, GPR120 is described as a switch governing the bi-potential fate of bone marrow MSCs versus osteogenesis or adipogenesis [129,135]. Gao et al. first showed that high-dose stimulation with antagonist TUG-891 at micromolar concentrations triggered GPR120-mediated ERK1/2 signaling, upregulating osteogenesis markers in MSCs, whereas low-dose activation of GPR120 favors fat differentiation due to p38 MAPK onset [135]. This biphasic behavior highlights that the receptor’s signaling output is not linear and may depend on ligand concentration, receptor internalization, or differential engagement of β-arrestin vs. G-protein branches. Such complexity could explain inconsistent findings among in vitro and in vivo studies where has been showed that treatment with a high-dose GPR120 agonist in osteoporotic mice effectively prevented bone loss, supporting the receptor’s pro-osteogenic role when robustly activated [136].

Beyond differentiation, GPR120 is also well characterized for its anti-inflammatory action in innate immune cells, particularly macrophages, where a switch to an anti-inflammatory phenotype and suppression of NF κB signaling were observed upon omega-3 binding [137,138]. A landmark study by Oh et al. [137] demonstrated that stimulation of GPR120 in RAW 264.7 cells and primary mouse macrophages suppresses the production of pro-inflammatory cytokines (e.g., TNF-α, IL-6) in response to lipopolysaccharide (LPS). Mechanistically, this effect is mediated by β-arrestin2-dependent inhibition of the TAK1–IKK–NF-κB signaling cascade, which prevents the transcription of inflammatory genes [137]. Furthermore, GPR120 activation in macrophages not only reduces inflammation but also induces a phenotypic switch from M1 to M2 macrophages, remarked by upregulation of anti-inflammatory markers such as IL-10 and Arg-1 [137]. This M2 polarization is critical for tissue repair and resolution of inflammation, indicating that GPR120 is a key regulator of immunological tone in damaged tissues. Nonetheless, direct casual links between MSCs-specific GPR120 signaling and sustained tissue regeneration remain limited. There is growing rationale to propose that similar anti-inflammatory pathways may operate in the MSC microenvironment, where MSCs interact closely with immune cells and are themselves responsive to lipid signals. MSCs are known to express GPR120 [131], and preclinical models suggest that the activation of GPR120 in MSCs can alter their cytokine secretion profile, enhance survival, and modulate differentiation, particularly under inflammatory or injury conditions. For example, Gao and coworkers reported that EPA treatment of bone marrow-derived MSCs upregulated autophagy and reduced apoptosis via a GPR120-mediated mechanism, suggesting direct intracellular effects in addition to environmental modulation [139]. Still, whether these mechanisms are sufficient to confer durable therapeutic improvement in vivo remains uncertain, since excessive GPR120 activation could theoretically blunt necessary early immune responses or shift metabolism toward lipid dependence at the expense of matrix synthesis.

Together, these findings support the concept that GPR120 may act as a lipid-sensing immunometabolic regulator in MSCs, integrating signals from dietary or local omega-3 FA to fine-tune the cells immunoregulatory and regenerative functions. By inhibiting NF-κB and engaging pro-survival and differentiation pathways such as PI3K/AKT, GPR120 may help MSCs maintain a reparative phenotype in inflammatory environments—an especially relevant feature in musculoskeletal pathologies where chronic inflammation impairs regeneration. However, the dual function of GPR120 in both metabolism and inflammation control suggests that its therapeutic potential depends on careful calibration of ligand type, dose, and timing. Future work should disentangle whether GPR120 primarily promotes regeneration through intrinsic MSC programming or via crosstalk with macrophages and chondrocytes within the joint niche.

#### 4.2.3. Toll-like Receptor 4 (TLR4)

TLR4 is an innate immune receptor known to detect bacterial LPS, as well as saturated FA such as palmitic acid and stearic acid, and endogenous danger-associated molecules like HMGB1, HSP60 and fibronectin fragments [140,141,142,143]. These ligands activate TLR4-mediated inflammatory signaling, contributing to chronic inflammation and tissue damage in various pathological contexts including PTOA. MSCs express TLR4 and its activation serves as an inflammatory trigger that can dramatically change MSC behavior. Whereas resting MSCs or those stimulated via receptors like TLR3 (which is sparked by viral RNA (dsRNA)), exhibit immunosuppressive properties. TLR4 engagement induces a shift toward a pro-inflammatory phenotype, a process described as MSC polarization. Recent research of Kaundal et al. illustrates this dichotomy [144]. Specifically, when human MSCs were primed with a TLR3 agonist (poly I:C), the MSC-2 phenotype associated with anti-inflammatory and immunosuppressive functions is favored. They observed upregulation for IL-10 and G-CSF production and promotion of Treg expansion, leading to effective suppression of effector T cell responses, associated with anti-inflammatory and immunosuppressive functions [144]. These data highlight that TLR-driven polarization is not a fixed property but strongly dependent on the nature, intensity, and timing of stimulation; an important factor when translating these findings into in vivo or chronic disease conditions. Conversely, MSCs stimulated with TLR4 ligands (e.g., LPS) increased the expression of pro-inflammatory mediators like IL-8, CXCL10 and CXCL12, initiated the NF-κB nuclear translocation and failed to support Treg induction, promoting the MSC-1 stated [144]. Notably, the authors pointed out that TLR4-MSCs not only secreted pro-inflammatory factors but may also impair the therapeutic efficacy of MCSs in some inflammatory pathological contexts. These findings suggest that TLR4 acts as an inflammatory toggle, tipping MSCs from helpers to potentially hinderers of regeneration. This has important implications, particularly in chronic inflammatory diseases where MSCs may be persistently exposed to TLR4 ligands (for example, damage-associated molecules in an arthritic joint), pushing them into a dysfunctional MSC-1 state. However, the extent of this impairment remains debated: transient or low-level TLR4 activation may actually prime MSCs for more balanced immune responses, while sustained or repeated engagement tends to drive dysfunction. This duality suggests that TLR4 signaling is contextually adaptive rather than inherently damaging.

Indeed, chronic low-grade activation of TRL4 is one hypothesis of MSC impairment in obesity and type 2 diabetes, where elevated free FAs and endotoxin can engage TRL4 and create a vicious cycle of inflammation [145]. Moreover, TLR4 signaling in MSCs has been shown to alter their differentiation potential, impairing osteogenesis toward a fibro-inflammatory phenotype, which ultimately compromises effective tissue repair. In vitro exposure of bone marrow MSCs to LPS leads to upregulation of pro-inflammatory cytokine IL-6 and IL-1β, despite concurrent activation of Wnt3a/Wnt5a-mediated osteogenic differentiation pathways [146,147]. In parallel, DAMP mediated TLR4 activation further disrupts the balance between inflammation and bone formation, exacerbating regenerative dysfunction [148]. This coexistence of inflammatory and osteogenic cues under TLR4 stimulation may reflect a compensatory but inefficient repair attempt, rather than a complete suppression of regeneration. Nevertheless, given that most of these insights are derived from acute stimulation models, it remains uncertain how well they capture the adaptive responses of MSCs under chronic exposure, such as in human OA or diabetes-associated bone loss.

On the other hand, blocking TLR4 or modulating its pathway can rejuvenate MSCs therapeutic properties, restoring their immunomodulatory capability. Emerging studies on preclinical models demonstrated that targeting the TLR4–NF-κB axis using bioactive compounds like polyphenols and selective TLR4 antagonists (i.e., Eritoran (E5564), Resatorvid (TAK-242), M62812, and small molecule C34) can effectively mitigate the pro-inflammatory shift and preserve MSCs therapeutic function under inflammatory conditions [149]. However, complete suppression of TLR4 signaling may blunt innate sensing and reparative crosstalk with immune cells. Therefore, achieving partial modulation rather than total inhibition could be more physiologically beneficial.

In summary, TLR4 on MSCs is a double-edged sword: while it is essential for sensing danger signals as part of the host-defense sensing machinery, its activation during tissue regeneration tends to dampen MSCs’ immunosuppressive abilities and foster chronic inflammation. The challenge, therefore, lies in distinguishing between transient activation necessary for immune orchestration and chronic signaling that perpetuates MSC dysfunction. This understanding has led to strategies where MSCs are preconditioned with specific TLR ligands to skew their phenotype. For instance, transient stimulation through TLR3 can “license” MSCs for therapeutic use by enhancing IL-10 and IDO, whereas avoiding TLR4 activation is desirable for maintaining MSCs in a healing mode. Moving forward, it will be essential to determine whether TLR4 inhibition should be systemic, localized, or temporally controlled, as each approach could have markedly different consequences for joint homeostasis. Controlling the TLR4-mediated inflammatory transition is therefore critical in chronic musculoskeletal diseases. In fact, in contexts like PTOA, therapeutic strategies that block TLR4 signals within the joint could help prevent MSCs and resident immune cells from perpetuating inflammation long after an injury.

Overall, the modulation of MSCs through lipid pathways represents a convergence of metabolic and regenerative medicine (Table 1). It suggests that drugs targeting these receptors could be repurposed to enhance musculoskeletal regeneration. For instance, some of them are already in use for metabolic diseases and selective PPAR-γ inhibitors or GPR120 agonists might be used to improve bone density in OP or to augment the efficacy of MSC-based therapies in joint repair. This theme underscores a sophisticated view of MSCs not just as building blocks for tissue, but as responsive “immunomodulatory factories” whose function can be tuned by lipid signals in their niche.

## 5. Key Themes and Trends

The last few years have witnessed a transformative conceptual shift from the classic paradigm of structural repair, where cells, biomaterials and growth factors are the main players, to high-resolution technologies, to recognizing immunometabolism not only as a downstream effector of inflammation but as a master regulator of regenerative outcomes. This sets the stage for decoding why lipid metabolism matters for tissue regeneration and how it can be harnessed therapeutically through advanced novel perspectives. Furthermore, traditional musculoskeletal therapies have focused on systemic suppression of inflammation (e.g., NSAIDs, corticosteroids, anti-TNF agents) [150]. In contrast, current trends emphasize precision modulation of lipid signaling pathways including PPARγ agonists, GPR120 ligands, FA uptake via CD36, and β oxidation pathways that can tip macrophage and MSC polarization toward pro regenerative phenotypes [151,152]. This paradigm reframing reflects a broader revolution taking hold across musculoskeletal regenerative medicine, transitioning from descriptive studies to mechanistically and clinically actionable strategies and opening up more strategic, system-level approaches looking at inflammation not as a problem to suppress, but as a process to guide and recalibrate metabolically. In this context, the integration of advanced high-resolution imaging with advanced modeling systems (e.g., organs-on-chip) and targeted engineering of biomaterials with specific lipid compositions represent powerful avenues for next-generation therapies. Ultimately, regenerative immunologists and scientists must embrace not only the identity of biologists but also that of designers, leveraging sophisticated understanding of lipid immunometabolic circuits to engineer solutions that precisely target pathological states and accelerate regeneration.

### 5.1. Technological Innovations

In this context, a newfound enthusiasm has coalesced around sophisticated high-throughput technologies now capable of resolving thousands of individual molecular species from minimal sample volumes, paving the way toward eventual single-cell resolution [153]. When integrated with spatial imaging, such as MALDI-based mass spectrometry imaging, secondary ion mass spectrometry, and Raman-based metabolic imaging, researchers can map lipid gradients in situ across cartilage or bone tissue, linking spatial lipid distribution to local inflammation or regenerative zones [154]. These methods can offer valuable insights into the profound cellular heterogeneity with unprecedented resolution underpinning lipid metabolism within MSCs and immune cells alike. For the first time, it can be visualized how localized lipid gradients dictate MSCs differentiation pathways or drive macrophages toward regenerative, anti-inflammatory states, resolving questions that traditional bulk analyses have consistently failed to address.

### 5.2. Organoids, Organ-on-Chip, and Metabolic Precision

Alongside these analytical breakthroughs, an exciting trend is the advent of microfluidic organ-on-a-chip models and immuno-metabolic organoids [155,156,157]. By recapitulating in vitro the complexity of tissue-specific niches, mimicking both mechanical and immune interaction via perfusion gradients, controlled inflammatory stimuli, and metabolite exchanges, these platforms can transform immunometabolic studies. They offer scalable and powerful tools not just for modeling tissue process but also rigorous testing concepts involving lipid signaling modulation and allowing real-time interrogation of how MSCs respond to defined lipid milieus under physiological relevant mechanical stress or simulated injury [155]. Such microphysiological systems promise to bridge a long-standing translational gap, allowing the precise dissection of how lipid mediators influence immune-stromal crosstalk, inflammation resolution, and tissue repair processes in dynamically controlled conditions reminiscent of the native niche.

### 5.3. Combination Therapies: Lipids Meet Biomaterials and Extracellular Vesicles (EVs)

A very emerging direction relies on the strategic coupling of lipid mediators with biomaterials or EVs in order to move from the endogenous production to “kick in” during healing. The goal is not limited to passive release but focalized to the orchestrated crosstalk [158]. In this scenario is not just a matter of immunomodulation, we can speak about “immuno-choreography”. Engineered biomaterials such as smart hydrogel used as lubricants and osteoinductive scaffolds can be loaded or functionalized with pro-resolving lipid mediators to enable spatial and temporal control on delivery and lipid signaling dynamics, guiding immune and MSC behavior during tissue restoration [76,159,160]. New generations could be far smarter: they can tune release kinetics in response to enzymatic cues from inflammation microenvironment or pH changes from osteoclast activity or even respond dynamically to mechanical strain. Their potential becomes more extensive when EVs are involved. EVs derived from MSCs or macrophages naturally carry lipid cargo [158], but interestingly, this can be enriched with specific payload like oxylipins and SPMs to deliver immunometabolic cues with high fidelity and minimal off-target toxicity [161]. The potential of synergy makes these approaches especially promising, transcending monotherapy. Lipids with biomaterials create permissive niches, EVs fine-tune the local immune dialect; together, they generate a microenvironment that “remembers” how to regenerate. This opens the door to constructing programmable therapeutic platforms able to deliver a cargo as well as adapting in real time to the tissue’s evolving immunometabolic needs.

### 5.4. An Emerging Vision: Personalized Regeneration

In some clinical milieu, there is continuous interest in patient-specific strategies grounded in biological profiling. Personalized regenerative medicine has long focused on cellular compatibility and biomechanical matching [162]. Additionally, its combination with machine learning could allow orthobiologics to be tailored injury type and in turn patient’s systemic lipid tone [163]. The integration of lipidomic signatures can introduce a patients’ immunometabolic fingerprint, making a future where joint and bone repair can be guided by a patient’s lipid immunophenotype increasingly plausible. Lipidomic profiling of impaired tissues could predict the responsiveness to therapies with remarkable accuracy, enabling predictive lipid signatures to serve as biomarkers for precise therapeutic matching [164]. However, personalized medicine implies also adaptation. Future regenerative strategies may be stratified by anatomical site or disease stage with the lipid reactivity of the patient’s immune system [162]. Imagine tailoring MSC priming protocols based on the patient’s serum oxylipin profile or adjusting scaffold composition to match local prostaglandin synthesis capacity. This is a sign of an approaching era where orthobiologics are metabolically bespoke. In this vision, lipid signaling is a diagnostic compass and a design variable, allowing scientists to move from reactive to responsive medicine. And critically, it demands that we rethink our clinical endpoints: no longer just “pain relief” or “bone fill,” but reestablishment of immunometabolic homeostasis within the regenerating tissue. The implications suggest that regeneration is a metabolic negotiation above merely a cellular task; a conversation between cell, niche, and system, mediated in part by lipid signals. The better we learn to listen and speak that language, the more fluently we can guide tissue back to health.

In summary, the stage is set for lipids to transform from passive metabolites to active protagonists into an unfolding scientific narrative poised to revolutionize regenerative medicine in the years ahead. Indeed, the field is transitioning from descriptive studies to actionable strategies where the lipid-driven immunometabolism is deeply embedded in the logic of regeneration. The challenge now is to harness its complexity without oversimplifying and to translate this knowledge into therapies respecting the sophisticated biology that only beginning to understand. For musculoskeletal medicine, this path holds genuine promise: precision-targeted immunometabolism to rebuild bone and muscle rather than merely treating symptoms.

## 6. Clinical Applications and Translational Perspectives

This lipid-centric view is rapidly reshaping regenerative strategies, emphasizing cell-free, minimally invasive, and off-the-shelf approaches that are more readily translatable. However, despite the compelling body of evidence supporting the role of lipid-driven immunometabolism in modulating MSC function, its clinical application in musculoskeletal regeneration remains unexplored. While earlier sections of this review have detailed the underlying molecular pathways and emerging preclinical strategies, the transition from mechanistic insight to therapeutic implementation demands critical evaluation. In this section, we adopt a translational lens to reflect on the real-world feasibility of applying lipid-modulated MSC-based therapies in the clinical management of PTOA, OP, and related conditions.

Turning the understanding and biological potential of enhancing immunomodulatory capability of MSCs into a reliable and reproducible therapeutic strategy requires overcoming several key challenges including the dynamic regulation of lipid metabolism in vivo, variability in MSC sources, the delivery and stability of bioactive lipids, and the lack of predictive preclinical models. Moreover, from a regulatory standpoint, lipid-based modulation introduces novel variables in cell therapy manufacturing and safety assessment that remain insufficiently addressed in current guidelines.

### 6.1. Translational Hurdles

Several scientific and logistical barriers exist on the path to this concept becoming a clinical reality. One of the most pressing issues is the complexity of in vivo lipid signaling regulation. While in vitro studies are significant and promising, the precise and safe targeting of lipid metabolic pathways and reproducing these effects consistently in vivo remains unpredictable, especially within inflamed or metabolically dysregulated environments. Translating findings from in vitro models to in vivo applications remains challenging due to the complex and dynamic microenvironment in living organisms, which cannot be fully replicated in vitro [165]. Factors such as cellular interactions, immune context, and systemic signaling influence MSC behavior and therapeutic efficacy in vivo, underscoring the need for cautious interpretation of in vitro results and the development of more predictive preclinical models [165]. Moreover, current animal models often inadequately mimic the complex interaction between PTOA and OP pathophysiology and metabolic scenario [166]. This lack of robustness of in vivo models limits the predictive value of preclinical studies and slows translation. There is a need for more predictive and integrative in vivo platforms, as well as emerging technologies such as tissue-on-chip systems, which could help evaluate the efficacy and safety of lipid-modulated MSC therapies in a controlled, yet biologically relevant, setting. Furthermore, significant species-specific differences exist in lipid metabolism, which pose challenges in translating findings from animal models to humans [167]. Variations in lipid composition, biosynthesis pathways, and metabolic regulation across species may impact the relevance and applicability of preclinical musculoskeletal research, necessitating cautious interpretation when extrapolating results to human biology.

On the other hand, lipid mediators are highly bioactive (particularly oxylipins and DHA derivates), inherently unstable and pleiotropic. Despite most SPMs exhibiting tissue-specific and self-limiting effects with favorable safety profiles in preclinical investigations, some studies have reported that their systemic modulation can in some contexts lead to unintended off-target effects that complicate clinical applicability [91,92]. This challenge often arises from receptor crosstalk, oxidative byproducts, or supra-physiological concentrations [168] and thus the overall outcome depends on local receptor expression, metabolic turnover, and the inflammatory milieu rather than intrinsic toxicity. In particular, free lipids injected into the joint or bone cavity are rapidly degraded or cleared depending on local concentration, enzymatic activity, and oxidative stress, which limits their therapeutic persistence and efficacy. Therefore, advanced and intelligent delivery strategies are critically needed to protect, control release, and enhance localization of lipid mediators for improved clinical outcome. In this scenario, bioactive scaffolds or injectable hydrogels and nanoparticle formulations capable of sustained, localized release in the tissue microenvironment represent essential next-generation technologies.

Another layer of complexity arises from donor viability and MSC sources [169]. Formulations derived from adipose stromal fractions, particularly the infrapatellar fat pad-derived MSCs, rich in lipid mediators, are gaining traction in clinical trials as promising therapeutic options for OA and related degenerative diseases [16]. While attractive due to their accessibility and immunomodulatory capacity, they can display significant heterogeneity in their lipid profiles depending on the donor’s age, sex, metabolic status, and comorbidities. This variability significantly hampers the standardization of therapeutic products, reproducibility and scaling for clinical use, requiring the development of quality control frameworks possibly including lipidomic profiling as a batch release criterion.

Finally, safety and regulatory concerns come into play. Bioactive lipids can exhibit dose-dependent biological effects and, if administered improperly, they risk triggering unintended pro-inflammatory responses or systemic adverse events. For that reason, comprehensive preclinical dose–response studies and toxicology assessments alongside meticulous clinical dose optimization are crucial early in development. Early and rigorous regulatory evaluation must also be navigated carefully to mitigate risks and guide safe clinical translation. The regulatory authorities necessitate clear definitions of manufacturing processes, standardization of therapeutic formulations, and detailed characterization of the bioactive components, ensuring consistency across production batches and clinical trials. Engagement with regulatory bodies, such as the U.S. Food and Drug Administration (FDA) or the European Medicine Agency (EMA), beforehand will facilitate clear guidance and align preclinical development strategies with regulatory expectations, thus smoothing the transition toward clinical application.

### 6.2. Key Requirements for Clinical Translation

Addressing these hurdles will require a focused, coherent strategy that engages multidisciplinary efforts between regenerative medicine, lipid biology, bioengineering, and clinical science.

One promising avenue lies in the development of lipid-informed quality metrics for MSC products. Therefore, establishing a standardized lipid profile for MSCs under varying physiological and pathological conditions is paramount. This would ensure uniformity in therapeutic preparations, greatly enhance predictability and reproducibility in clinical outcomes. In parallel, leveraging lipid fingerprinting as a companion diagnostic tool could profoundly impact translational efforts. In this context, lipidomic profiling could be employed as a tool for characterization and, most importantly, as a predictive biomarker for therapeutic potency and donor suitability. In this way, clinicians and researchers can reliably identify optimal MSC sources thus significantly reducing variability and improving efficacy.

On the delivery side, translating these complex therapies into practical clinical interventions demands innovation in technologies. Biomaterial-based systems, such as sophisticated hydrogels or bioactive scaffolds designed to simultaneously deliver MSCs and bioactive lipids while also providing a matrix for cell retention and mechanotransduction, can offer promising solutions. These platforms must guarantee controlled release, structural stability, and precise localization, essential for maximizing therapeutic benefits while minimizing side effects. They can also be co-designed with anti-fibrotic or immunoregulatory cues to further enhance therapeutic outcomes.

Moreover, integration with precision medicine frameworks may enable more personalized approaches, matching patients with specific lipid imbalances or inflammatory signatures to tailored MSC-lipid constructs. This will likely require the development of companion diagnostics, including synovial fluid or serum lipid profiling, to stratify patient populations and monitor therapeutic responses. The development of patient-specific lipid fingerprints could guide the selection of the most appropriate therapeutic strategy, whether based on autologous MSCs, off-the-shelf products, or cell-free approaches. Technologies such as bioprinting, multi-omics integration, and AI-assisted design will likely accelerate this vision, providing tools to simulate, predict, and optimize therapeutic responses in silico before entering the clinic.

Collaborative efforts with regulatory bodies and participation in translational consortia will also be key to establishing standardized manufacturing protocols, release criteria, and safety monitoring tools, especially as lipid engineering introduces new variables in MSC therapy production.

Finally, perhaps the most critical factor in translating lipid-driven immunomodulation into clinical reality is the promotion of robust, multidisciplinary collaboration. Engaging a broad spectrum of expertise (e.g., engineers, rheumatologists, orthopedic surgeons, immunologists, and metabolic biologists) will drive integrated preclinical and clinical research, fostering rapid and effective resolution of challenges and significantly streamlining the path from bench to bedside.

### 6.3. Vision of the Orthopedic Future: Towards Lipid-Guided Immunometabolism and Regenerative Therapies

The clinical management of multifactorial and widespread musculoskeletal conditions such as PTOA and OP requires scalable, standardized products that are both effective and practical rather than tailoring therapies to individual patients. Orthopedic practices thrive on reliability and look forward to a more feasible strategy that relies on designing robust interventions that respond dynamically to local pathological cues, such as inflammation or metabolic dysregulation, thus maximizing therapeutic potential across patient populations. Clinicians who treat PTOA, OP or delayed union require standardized, reliable, and effective options. Such treatments ideally come as sterile and off-the-shelf products that perform predictably across diverse patient populations and clinical settings. The translational challenge, therefore, is to develop products that act as standardized hardware but think like biologics once inside the body. This is also the essence of the regulatory concept “scalable but targeted”. A practical route begins with harmonized allogenic MSCs manufactured under closed, Good Manufacturing Practice (GMP)-grade protocols to ensure batch-to-batch reproducibility. Additionally, potential pre-licensed lipidic preconditioning that fixes a pro-resolving secretome for the long haul can be performed before cryostorage, allowing release with a tight lipid fingerprint while decoupling potency from donor variability. At the point of care, the thawed cells are sealed inside a delivery platform whose chemistry is wired to local cues: ROS-cleavable linkers snap open in inflamed tissues [170], while pH-responsive micelles bleed lipids precisely where acidity betrays catabolic cartilage [171]. Such off-the-shelf single units packaged as a Class III combination product reduces surgical steps and aligns with regulatory framework. Moreover, because the fingerprint is batch-bound and not patient-specific, the lipidomic assay doubles as a potency test and simplifies regulatory filings, serving as a release criterion. Each lot is released only if it hits a predefined bioactive lipid range for which efficacy is validated in animal studies or in organ-on-chip pathological models and early human data.

Understandably, early and iterative conversation with the regulatory programs such as FDA’s INTERACT and EMA’s PRIME are ideal to vet needs and claims, ensuring that every adaptive feature agrees to a measurable surrogate endpoint as biomarkers, potency assays, and post marketing commitments. Meanwhile, parallel health-economic modelling demonstrates how off-the-shelf constructs shorten operating time, reduce rehab visits, and delay costly arthroplasty, while also streamlining regulatory review and expediting reimbursement decisions by health systems, thereby improving their clinical adoption.

In conclusion, the bridge to clinic is built from uniform building blocks endowed with situational intelligence: standardization earns the surgeon’s trust; understanding and controlling lipid effects earns the tissue’s respect. Only by fulfilling both criteria can lipid-driven immunometabolic therapies successfully transition from experimental study to established orthopedic clinical practice.

## 7. Conclusions

The advancements discussed in this review highlight the importance of understanding the mechanisms behind lipid-driven immunometabolism as well as the critical translational pivot toward lipid-informed regenerative solutions, underscoring the potential for transformative therapies in the next years. In the near future, one can envision engineered injectable constructs that combine pre-conditioned MSCs with lipid-loaded hydrogels or EVs, designed to respond to the local immunometabolic environment of the diseased joint or bone.

The long-term horizon still points toward precision through smart design rather than artisanal customization. While fully individualized regenerative therapies remain conceptually appealing, the field is shifting toward a more pragmatic model of precision medicine, where defined patient subgroups are matched with robust, scalable therapeutic platforms. The ultimate metric of success will not be how unique each product looks in vitro, but how consistently it restores mobility and dampens inflammation across the heterogeneous, aging musculoskeletal patient population. In this context, lipidomic profiling could serve as a stratification tool to identify patients who are more likely to benefit from immunometabolically tuned interventions. Additionally, the integration of this assay with immunometabolism and regenerative engineering sets the stage for a new generation of therapies capable of simultaneously controlling inflammation and promoting tissue repair in musculoskeletal disorders.

Achieving this vision requires sustained investment in cross-disciplinary academic consortia and industry partners, along with regulatory innovation and translational infrastructure. Prospectively, the creation and active participation of translational consortia (e.g., CEPI, NIH, CIRM, EIT Health) represent crucial mechanisms to bridge academic research with clinical application. Such networks leverage collective expertise, infrastructure, and resources, significantly enhancing translational efficacy and accelerating the journey towards effective, personalized lipid-driven therapies.

In conclusion, lipid-driven immunometabolic therapies represent a highly promising complex frontier that can graduate from innovative bench science to reliable operating room workhorses within the next decade. Addressing critical translational hurdles with a multidisciplinary, methodical, and visionary approach can pave the way for transformative therapies in musculoskeletal medicine. The opportunity to create more targeted, safe, and effective therapies for musculoskeletal conditions makes lipid-driven immunometabolism a field with extraordinary potential and one that is now ready to move beyond the bench.

## Figures and Tables

**Figure 1 ijms-26-10117-f001:**
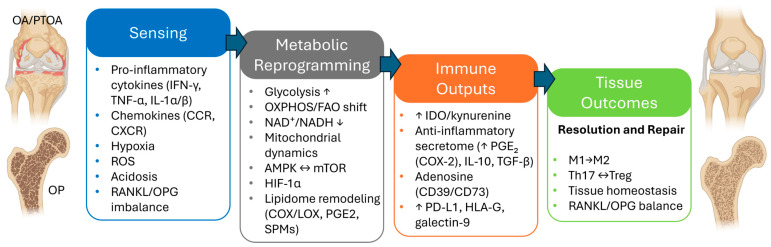
MSC plasticity and reprogramming in musculoskeletal disorders: a four-stage framework from sensing to resolution of inflammation and tissue repair. (Arrows notation: ↑ = increase; ↓ = decrease; ↔ = bidirectional, reciprocal regulation; → = shift toward the indicated state).

**Table 1 ijms-26-10117-t001:** Key lipid-driven immunometabolic pathways, sensors and regulators in MSCs, with predominant disease effects and candidate intervention points. (Arrows notation: ↑ = increase; → = shift to-ward the indicated state).

Regulator/Pathway	Core Role	Post-Traumatic Osteoarthritis (PTOA) Signature	Osteoporosis (OP) Signature	Therapeutic Leverage Points
Lipid mediators: PGE_2_ (AA → COX-2)	Context-dependent immunosuppression. [75,80,92]	Often ↑ with TNF-α/IFN-γ; helps curb synovitis by suppressing Th1/Th17; inhibit T cell and skew M2 macrophages. Excess may be catabolic. [35,75]	Supports anti-inflammatory milieu yet requires dosing control for bone remodeling. High systemic may impair bone. [75]	Time/dose-controlled COX-2/PGE_2_ tuning. [92]
ω-3 PUFA (DHA/EPA)	Increases biosynthetic/metabolic activity; enriches therapeutic secretome. [96]	Enhances immunosuppressive and regenerative potency in inflamed joints. [96]	Helps correct chronic inflammation that accelerates bone loss. [97]	Ex-vivo DHA/EPA priming; systemic supplementation (pilot human data). [91,96]
SPMs (RvE1, RvD1, MaR1, lipoxins)	Pro-resolving, pro-repair. [85,86,87]	M1→M2, Th17→Treg; intra-articular SPMs tone protects cartilage and subchondral bone; MaR1 prevents PTOA in models. [90]	Counteracts low-grade inflammation; can reduce osteoclastogenesis and support osteoblasts. [97]	ω-3/SPM supplementation or delivery (free, biomaterial or EV-loaded). [85,86,87,91]
PPARγ	Anti-inflammatory lipid-sensing transcriptional control. [99,105,106,107,108]	Overactivation may blunt osteochondral repair. [113]	↑ PPARγ drives MSC adipogenesis at expense of osteogenesis; bone loss. [100,101]	Selective PPARγ antagonists in OP. [101]
PPARβ/δ	Chondroprotective lipid-sensing transcriptional control. [109]	Modulation can lessen joint inflammation and improve MSC chondrogenesis; loss-of-function enhances MSC immunosuppression via glycolytic shift. [110,111,112,113,114,115,116]	Indirect support of osteogenesis via oxidative metabolism. [109]	Context-specific agonism/antagonism depending on cartilage-immune targets. [114,115,116]
GPR120 (DHA/EPA receptor)	Anti-inflammatory signaling; MSC fate control toward osteogenesis. [121,122,123,126,127]	Activation reduces NF-κB in chondrocytes and macrophages; preserves cartilage. [120]	Promotes osteogenesis and restrains adipogenesis; improves bone microarchitecture. [119,125]	Ex-vivo MSC priming with DHA/EPA or GPR120 agonists. [117,118,119,125]
TLR4 (DAMP/PAMP sensor; saturated FA-responsive)	Inflammatory toggle. [129,130,131,132,133]	Chronic activation sustains synovitis and impairs regeneration. [133,137]	Chronic activation skews marrow niche toward osteoclastogenesis. [134]	Antagonize TLR4 (e.g., TAK-242, Eritoran in preclinical); prefer TLR3 licensing for MSC-2 anti-inflammatory phenotype. [133,138]

## Data Availability

No new data were created or analyzed in this study. Data sharing is not applicable to this article.

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
