# Peer review of "Lipid-Driven Immunometabolism in Mesenchymal Stromal Cells: A New Axis for Musculoskeletal Regeneration"

_ijms, 2025, doi:10.3390/ijms262010117_

Round 1
Reviewer 1 Report
Comments and Suggestions for Authors
The article is reviewing emerging data on the role of the immune system in the bone and muscle tissue regeneration. In the manuscript, focus lies on the immunological signaling and its impact on cell metabolism, ‘immunometabolism’, encompassing a complex communication network between immune cells and muscle/bone cells - Mesenchymal Stromal Cells. In this network, lipid metabolism has prominent importance, since immunometabolism outcomes ultimately determine whether the tissue will be regenerated or slide into degenerative processes.
The article offers a comprehensive and inspiring overview of the recent scientific evidence, placing them into the clinical context. Special quality of this review is its interdisciplinary approach, tightly and carefully woven both into the structure and the logic of the manuscript. The authors have succeeded in bridging molecular, in vitro lab findings, with clinical trials, emphasizing the potential benefits for individual patients and clinicians. Future perspectives are extremely well depicted and analyzed; industrial, regulatory and economic issues included. It is a valuable contribution for the research community, clinicians and regulators.
Minor Issues
Line 11 - “immunometabolism—the cellular metabolism–immune system interplay”, would be clearer to write “immunometabolism (cellular metabolism–immune system interplay)”
Line 163 – - „OA“ abbreviation is used for the first time in the text, please include the full term
Line 927 - OA is missing from the list of abbreviations, as some others. Please extend.
Author Response
The Authors truly appreciate the reviewer’s positive and encouraging assessment and the helpful minor suggestions. We have implemented all requested revisions, expanded the Abbreviations list to include all terms used, and conducted a consistency audit across the manuscript. All edits are presented in tracked changes.

Reviewer 2 Report
Comments and Suggestions for Authors
Its a timely and comprehensive review explaining the effect of lipid metabolism in MSC biology.
However, the amount of information presented is large, so the reader would like to see one or more cartoons that explain the mechanisms and the outcomes. Some open questions remain to the reader. What lipids could modulate the physiological behaviour of MSCs and how?
If MSCs are preconditioned with lipids, what is to be expected and why?
Can these conditions happen in vivo?
Any future applications or experimental designs?
Author Response
We sincerely thank the reviewer for the insightful and constructive comments. We fully agree
that the investigation of lipid metabolism in mesenchymal stromal cell (MSC) biology is a highly
relevant and emerging topic. However, we would like to highlight that the field of lipid-driven
immunometabolism in MSCs is still in its infancy compared to the more established knowledge
in immune cells. While lipid regulation of immune cell function and metabolism has been
extensively described, the application of these concepts to MSCs—given their
immunomodulatory capacities—is relatively novel, with many mechanistic details yet to be
elucidated.
In particular, the identity of lipids capable of modulating MSC physiological behavior is currently
mostly inferred from immune cell studies and hypothesized classes of bioactive lipids, such as
polyunsaturated fatty acid derivatives. Detailed functional and mechanistic data on these lipids
in MSCs remain limited. Preconditioning MSCs with lipids are also still under investigation but
we are confident that will show promising effects on proliferation, differentiation, and
immunomodulatory functions, but the expected outcomes and underlying molecular pathways
require further efforts. Moreover, whether such lipid-mediated modulation occurs physiologically
in vivo remains an open question due to the complexity of the in vivo milieu and current scarcity
of in vitro direct evidence. Potential future clinical applications and related challenges were
implemented in section 6. Clinical Applications and Translational Perspectives and focused on
musculoskeletal scenario.
We acknowledge these open questions and believe they reflect a promising frontier in MSC
biology that warrants further research; although, we have carefully revised the manuscript and
incorporated clarifications along the text wherever necessary to address the valuable points
raised by the reviewer. Regarding the suggestion to include cartoons or schematic
representations illustrating the lipid-driven immunometabolism in MSC and their outcomes, the
Authors believe that the field remains currently too unexplored to propose a clear and definitive
schematic that could risk generating confusion. Instead, they consider that the provided table in
the manuscript offers a more concrete and accurate overview of the current known lipid
mediators and effects within MSC biology. Once again, we deeply appreciate the reviewer’s
valuable suggestions, which have helped us clarify the novelty and limits of the current
knowledge and better frame the discussion for our audience.

Reviewer 3 Report
Comments and Suggestions for Authors
This review paper comprehensively examines how lipid-driven immunometabolic pathways regulate mesenchymal stromal cell function in musculoskeletal regeneration, proposing that lipid mediators like specialized pro-resolving mediators (SPMs) and bioactive fatty acids can be strategically targeted to enhance MSC immunomodulation and tissue repair in conditions like post-traumatic osteoarthritis and osteoporosis. While the paper presents an innovative framework connecting lipid metabolism to regenerative medicine with strong translational potential, it requires revision for several typos, overly complex sentence structures, inconsistent terminology, and would benefit from adding a limitations section and clearer transitions between major conceptual sections.
Q1:Line 44: "Ket Themes" should be "Key Themes"
Q2:"Cellular Reprogramming to Unleash Immunomodulatory Function" - the phrase "unleash" is overly dramatic for a scientific paper. Consider "Cellular Reprogramming and Immunomodulatory Function"
Q3:Line 205: "impedess" should be "impedes"
Q4:Inconsistent terminology - The paper switches between "mesenchymal stromal cells" and "mesenchymal stem cells" - while MSC can refer to both, consistency would improve clarity. Choose one primary term and note the alternative early on.
Q5:line 296:"so call" should be "so-called"
Q6:"PPAR-γ antagonist (e.g. BADGE)" - While BADGE has antagonist properties, it's more accurately described as a partial antagonist/modulator. This nuance matters for interpretation.
Q7:The claim about "off-target toxicity" for lipid mediators needs qualification - some SPMs have shown off-target effects in specific contexts, but others have excellent safety profiles.
Q8:Section 4.1 title repetition - The section is titled "MSC Secretome, Lipidome and Bioactive Lipid Mediators" but subsection 4.1 has the identical title. This is confusing. Consider renaming 4.1 to something more specific like "SPMs and Omega-3/6 Fatty Acid Derivatives in MSC Function"
Q9:"cannot gamble on bespoke" – what does this mean?
Q10:"find a permanent address on the orthopedic shelf" - what does this mean?
Q11:Some citations use ranges [1-7] while others list individually. Standardize.
Q12: "inflammation must be tightly controlled to permit healing rather than fibrosis" needs citation
Q13: "immunochoreography" please provide citations
Q14:PGE2 discussed multiple times without clear progression.
Q15:No discussion of limitations - The review should acknowledge: 1. Variability in MSC sources and donors. 2. Challenges in translating in vitro findings to in vivo. 3. Species differences in lipid metabolism
Q16: Limited discussion of dose-response relationships
Comments on the Quality of English Language
The English could be improved to more clearly express the research.
Author Response
The Authors truly appreciate the reviewer’s thoughtful and constructive comments. We have carefully revised the manuscript as suggested and we have also integrated discussions of limitations, open debates, and remaining knowledge gaps throughout the manuscript to enhance clarity and critical perspective. All edits are presented in tracked changes.
Q1:Line 44: "Ket Themes" should be "Key Themes"
The Authors thank the reviewer observation and have corrected the typo accordingly.
Q2:"Cellular Reprogramming to Unleash Immunomodulatory Function" - the phrase "unleash" is overly dramatic for a scientific paper. Consider "Cellular Reprogramming and Immunomodulatory Function"
The Authors appreciate the reviewer’s feedback regarding the use of the term “unleash,” which could be interpreted as overly dramatic. To reflect a more objective and academic tone while maintaining the intended scientific meaning, we have revised the title to: “MSC Plasticity in Musculoskeletal Disease: Cellular Reprogramming Underlying Immunomodulatory Function.” This phrasing conveys the mechanistic link between reprogramming and immunomodulatory activity without overstatement.
Q3:Line 205: "impedess" should be "impedes"
The Authors thank the reviewer for noting this typo. The sentence has been rephrased to improve clarity and correctness.
Q4:Inconsistent terminology - The paper switches between "mesenchymal stromal cells" and "mesenchymal stem cells" - while MSC can refer to both, consistency would improve clarity. Choose one primary term and note the alternative early on.
The Authors appreciate the reviewer’s insightful comment. To ensure consistency and align with current recommendations from the International Society for Cell & Gene Therapy (ISCT), we have standardized the terminology throughout the manuscript to “mesenchymal stromal cells (MSCs).” We also clarify early in the text that the term “mesenchymal stem cells” is commonly
used in the literature; however, we adopt “stromal” to more accurately reflect the heterogeneous, supportive, and immunomodulatory nature of these cells rather than a strict stemness definition.
Q5:line 296:"so call" should be "so-called"
The Authors thank the reviewer for identifying this typographical error. The phrase has been corrected to “so-called” in the revised manuscript.
Q6:"PPAR-γ antagonist (e.g. BADGE)" - While BADGE has antagonist properties, it's more accurately described as a partial antagonist/modulator. This nuance matters for interpretation.
The Authors thank the reviewer for this helpful clarification. We have revised the text to describe BADGE as a PPARγ modulator rather than a pure antagonist, reflecting its partial antagonistic activity and improving interpretative accuracy.
Q7:The claim about "off-target toxicity" for lipid mediators needs qualification - some SPMs have shown off-target effects in specific contexts, but others have excellent safety profiles.
The Authors appreciate the reviewer for highlighting the need to qualify the statement on potential off-target effects. We have revised the text to clarify that such effects typically reflect receptor cross-talk, oxidative modification, or exposure to non-physiological concentrations, rather than inherent toxicity. Supporting references have been added to substantiate both the mechanistic basis of these observations and the generally favorable safety profile of SPMs reported in translational studies.
Q8:Section 4.1 title repetition - The section is titled "MSC Secretome, Lipidome and Bioactive Lipid Mediators" but subsection 4.1 has the identical title. This is confusing. Consider renaming 4.1 to something more specific like "SPMs and Omega-3/6 Fatty Acid Derivatives in MSC Function"
The Authors recognize the potential confusion arising from the identical section and subsection titles. This was an inadvertent oversight during manuscript formatting and has been corrected to improve clarity and alignment with sections’ content.
Q9:"cannot gamble on bespoke" – what does this mean?
The Authors thank the reviewer for highlighting the potential ambiguity of this phrase. We have revised the text to clarify that clinicians require proven, standardized, and dependable treatment options rather than taking risks with individualized, rapidly made preparations that may have unpredictable outcomes or insufficient evidence of efficacy. This improvement should better convey the urgency for reliable therapies in clinical practice and aligns with the need for treatments that perform consistently across diverse patient populations.
Q10:"find a permanent address on the orthopedic shelf" - what does this mean?
The Authors sincerely thank the reviewer for the insightful comment regarding this phrase. To clarify, this metaphor signifies the transition of lipid-driven immunometabolic therapies from experimental research ("bench") to established, standardized treatments routinely used and trusted in orthopedic clinical practice (“the orthopedic shelf”). We have revised and refined this concept in the manuscript to enhance clarity and ensure that the discussion accurately reflects the importance of developing therapies that are both scientifically sound and clinically dependable.
Q11:Some citations use ranges [1-7] while others list individually. Standardize.
The Authors sincerely thank the reviewer for the comment on citation formatting. While we appreciate the importance of standardized citation styles, we would like to clarify that individual citations were used when referring to a specific study directly discussed or analyzed, as these require precise attribution. In contrast, citation clusters or ranges were employed when supporting broader topics, concepts, or statements with multiple related references, which helps improve readability and reduce excessive fragmentation of evidence. This approach balances clarity, conciseness, and accuracy, enabling readers to appreciate the breadth of supporting literature while clearly identifying key individual studies where needed. Although we have carefully reviewed the references to ensure accuracy and consistency throughout the manuscript.
Q12: "inflammation must be tightly controlled to permit healing rather than fibrosis" needs citation
The Authors are grateful to the reviewer for the insightful and constructive suggestions on referencing and we have refined the section with pertinent references.
Q13: "immunochoreography" please provide citations
The Authors thank the reviewer for the suggestion. We intended "immunochoreography" as a metaphorical expression to emphasize that in this emerging field, the immune response is not simply modulated passively but involves a highly coordinated and dynamic orchestration of cellular and molecular interactions, similar to a choreographed performance. To clarify this, we have now used quotation marks around "immuno-choreography" to highlight its figurative meaning.
Q14:PGE2 discussed multiple times without clear progression.
The Authors appreciate the reviewer's observation regarding the discussion of PGE2. To clarify, we introduced PGE2 early in the text because it is one of the most extensively studied and well-characterized immunoregulatory lipid mediators in mesenchymal stromal cells (MSCs), serving as a key reference point to evaluate their immunomodulatory capacity. The brief discussion provides a background to ground the reader in established knowledge. However, to maintain focus on novel and emerging research directions in the proposed topic, especially those related to receptor-mediated mechanisms, the discussion on PGE2 was intentionally concise and broad rather than deeply detailed. We truly appreciate the suggestion and plan to offer a more detailed discussion and clearer progression on PGE2, as well as other bioactive lipid mediators, in future review articles dedicated specifically to lipid mediator biology in MSCs.
Q15:No discussion of limitations - The review should acknowledge: 1. Variability in MSC sources and donors. 2. Challenges in translating in vitro findings to in vivo. 3. Species differences in lipid metabolism
The Authors appreciate the reviewer’s insightful comments regarding the discussion of limitations. All the suggested points have been expanded in section 6.1 to better address these critical aspects and clarify this limitation in future revisions to better acknowledge the challenges in translating preclinical lipid metabolism data to clinical contexts.
Q16: Limited discussion of dose-response relationships
The Authors acknowledge the reviewer’s comment regarding the limited discussion on dose-response relationships. Given the relatively new nature of the topic and its ongoing investigations in the musculoskeletal field, a comprehensive statement on dose-response cannot yet be definitively made. Nevertheless, dose-responsive effects are discussed where data are available, as exemplified in the section on GPR120, which highlights key ligand dose-dependent mechanisms in mesenchymal stromal cells. We believe this balanced approach reflects the current state of literature while paving the way for more detailed dose-response analyses in future studies.

Reviewer 4 Report
Comments and Suggestions for Authors
This review gives a timely overview of how lipid-driven immunometabolism regulates mesenchymal stromal cell (MSC) function and affects musculoskeletal regeneration. It combines discussions on lipid signaling, metabolic changes, and potential clinical applications, creating an original link between basic science and regenerative therapies. However, the paper is mostly descriptive, with too much background and limited critical analysis. Adding summary figures and more examples from clinical or preclinical studies would make the manuscript stronger.
Major Comments
- The review is well written and thorough, but it mainly summarizes existing studies without enough critical discussion. The authors should compare opposing findings (for example, the different effects of PPARγ activation vs. inhibition, or TLR4 activation vs. suppression) and discuss current debates or knowledge gaps.
- Sections 2–3 (on skeletal tissue metabolism and MSC plasticity) are too long and move away from the main topic. These parts should be shortened, keeping the focus on lipid signaling and its role in MSC immune regulation and regeneration.
- The Clinical Applications section should be expanded with examples of ongoing preclinical or clinical studies that support the translational relevance of lipid-based MSC therapies.
Minor Comments
- Make sure all abbreviations (e.g., SPMs, FAO, OXPHOS, OP, PTOA) are defined the first time they appear and used consistently throughout.
- Check font style, size, spacing, and line breaks for consistency. Section 4.2.1 (lines 464–519) looks different from other parts.
- Add missing references for Section 4.1 (lines 353–377).
- Provide references in Section 5, especially where studies are mentioned without citation.
- Add supporting references for Section 5.3 (lines 696–708).
- Include citations for the clinical studies mentioned in Section 5.4.
Author Response
We truly appreciate the reviewer’s encouraging assessment and the helpful minor suggestions.
We have implemented all requested revisions, expanded the Abbreviations list to include all
terms used, and conducted a consistency audit across the manuscript. All edits are presented in
tracked changes in the revised manuscript.
Major Comments
1. The review is well written and thorough, but it mainly summarizes existing studies
without enough critical discussion. The authors should compare opposing findings (for
example, the different effects of PPARγ activation vs. inhibition, or TLR4 activation vs.
suppression) and discuss current debates or knowledge gaps.
The Authors thank the reviewer for this valuable comment. We have now incorporated a more
critical discussion throughout the revised version, explicitly comparing opposing findings and
highlighting current debates and knowledge gaps to strengthen the scientific impact of the
review.
2. Sections 2–3 (on skeletal tissue metabolism and MSC plasticity) are too long and move
away from the main topic. These parts should be shortened, keeping the focus on lipid
signaling and its role in MSC immune regulation and regeneration.
The Authors sincerely thank the reviewer for this valuable suggestion. Accordingly, Sections 2
and 3 have been thoughtfully shortened to maintain focus on the novel and central topic of lipid
signaling in MSC immune regulation and tissue regeneration. While condensing these sections,
we ensured to still provide a general overview of the musculoskeletal environment and the
immunomodulatory potential of MSCs for the convenience and clarity of the reader. This
approach balances brevity with necessary context, enhancing the manuscript’s overall clarity,
impact, and focus.
3. The Clinical Applications section should be expanded with examples of ongoing
preclinical or clinical studies that support the translational relevance of lipid-based MSC
therapies.
The Authors thank the reviewer for the constructive suggestion. We agree that expanding the
“Clinical Applications” section with concrete examples of preclinical and clinical studies will
strengthen the translational relevance of lipid-based MSC therapies for musculoskeletal
disorders. Although, despite clinical studies of MSC therapies in musculoskeletal diseases like
osteoarthritis and muscle injuries are steadily increasing, no one specifically focuses on lipid-driven immunometabolism of MSCs due to the novelty of this line of research. Nonetheless, this
field is recognized as an emerging and high-potential frontier for regenerative and
immunomodulatory therapies. In light of this, we have emphasized the translational promise of
lipid-driven MSC immunometabolism for musculoskeletal regeneration in the Clinical
Applications as follow “despite the compelling body of evidence supporting the role of lipid-driven immunometabolism in modulating MSC function, its clinical application in musculoskeletal
regeneration remains unexplored”. Additionally, we would like to highlight that, given the novelty
of this research area, to the best of our knowledge, only one study has specifically investigated
the role and mechanism of action of a specialized pro-resolving lipid mediator in the context of
musculoskeletal MSC therapies. In particular, we had included and discussed the recent study
by Leite et al. (2025), which characterizes the Maresin 1-LGR6 axis and its anti-inflammatory
and disease-modifying effects in a mouse model of post-traumatic osteoarthritis induced by
anterior cruciate ligament transection [Leite CBG, Fricke HP, Tavares LP, et al. Maresin 1-LGR6
axis mitigates inflammation and posttraumatic osteoarthritis after transection of the anterior
cruciate ligament in mice. Osteoarthritis Cartilage. 2025 Jul;33(7):861-873]. The few other
studies reported in the literature and specifically in the musculoskeletal context have explored in
vivo the receptors as described in Section 4.2, which have been expanded to include a critical
discussion. We believe these revisions enrich the translational relevance of our discussion and
underline the emerging therapeutic potential of lipid mediators in this field.
Minor Comments
1. Make sure all abbreviations (e.g., SPMs, FAO, OXPHOS, OP, PTOA) are defined the
first time they appear and used consistently throughout.
The Authors thank the reviewer for the observation and expanded the Abbreviations list to
include all terms used.
2. Check font style, size, spacing, and line breaks for consistency. Section 4.2.1 (lines 464–
519) looks different from other parts.
The Authors appreciate the reviewer's comment, and they adjusted the spacing of Section 4.2.1
for consistency.
3. Add missing references for Section 4.1 (lines 353–377).
4. Provide references in Section 5, especially where studies are mentioned without citation.
5. Add supporting references for Section 5.3 (lines 696–708).
6. Include citations for the clinical studies mentioned in Section 5.4.
The Authors are grateful to the reviewer for the insightful and constructive suggestions on
referencing. The “Key Themes and Trends” section adopted a forward-looking perspective,
outlining opportunities in a field that is still taking shape. While, to the best of our knowledge, no
studies yet directly integrate the precise combination we propose—highlighting the novelty of
the field—the reviewer’s comment was especially helpful. Accordingly, the Authors have
expanded and refined the section with pertinent references

Reviewer 5 Report
Comments and Suggestions for Authors
The manuscript by V Velur et al., entitled "Lipid-Driven Immunometabolism in Mesenchymal Stromal Cells: A New Axis for Musculoskeletal Regeneration”, examines how lipid-derived mediators and metabolic pathways contribute to immunosuppressive capacity of mesenchymal stem cells (MSCs). Building on compelling evidence supporting the role of lipid-mediated immunometabolism in modulating MSC function, the Authors evaluate the potential for clinical application of lipid-modulated MSC-based therapy to rebalance the inflamed microenvironment and promote musculoskeletal regeneration, with an emphasis on cell-free, minimally invasive, and easily translatable approaches. This review could be accepted for publication in IJMS. However, some minor shortfalls should be addressed.
Comments for Authors:
The terms "NETosis" (line 270) and "osteomax" (line 274) are niche. For the reader's convenience, I recommend providing explanations in the text.
Line 354-355: “In summary, lipid immunometabolism is the nexus of metabolic, immunity, and regeneration, serving as a fundamental component of MSC biology”. Please clarify whether you mean: “metabolic immunity” or “metabolism, immunity”?
Line 533-534: “Its mechanism of action is based on ligand-induced conformational changes …”. In the context of the paragraph, it's unclear what mechanism of action is being referred to?
Line 879-883: “Meanwhile, parallel health-economic modelling demonstrates how off the shelf constructs shortens operating time, reduces rehab visits, and delays costly arthroplasty by strengthening the clinical argument to payers, which accelerate reimbursement, a critical translational checkpoint often overlooked by the scientific teams.” In this context, the emphasis on “scientific teams” seems unfair and offensive. I recommend using the impersonal form.
Many abbreviations, including rarely used ones, are not deciphered either in the text or in the abbreviations section.
Author Response
We truly appreciate the reviewer’s positive and encouraging assessment and the helpful minor
suggestions. We have implemented all requested revisions, expanded the Abbreviations list to
include all terms used, and conducted a consistency audit across the manuscript. All edits are
presented in tracked changes.
Comments for Authors:
The terms "NETosis" (line 270) and "osteomax" (line 274) are niche. For the reader's
convenience, I recommend providing explanations in the text.
We thank the reviewer for pointing out the need to clarify niche terms. We have added brief, in-text explanations at the first mention of NETosis and osteomacs to improve readability for non-specialist readers, and revised the sentences for grammar and parallel structure.
Line 354-355: “In summary, lipid immunometabolism is the nexus of metabolic, immunity, and
regeneration, serving as a fundamental component of MSC biology”. Please clarify whether you
mean: “metabolic immunity” or “metabolism, immunity”?
The Authors thank the reviewer for the comment and have adjusted the typos in the text.
Line 533-534: “Its mechanism of action is based on ligand-induced conformational changes …”.
In the context of the paragraph, it's unclear what mechanism of action is being referred to?
We thank the reviewer for this clarification request. The intended referent was GPR120;
therefore, to avoid ambiguity, we specified the receptor and clarified that the downstream
signaling is GPR120-mediated upon DHA binding.
Line 879-883: “Meanwhile, parallel health-economic modelling demonstrates how off the shelf
constructs shortens operating time, reduces rehab visits, and delays costly arthroplasty by
strengthening the clinical argument to payers, which accelerate reimbursement, a critical
translational checkpoint often overlooked by the scientific teams.” In this context, the emphasis
on “scientific teams” seems unfair and offensive. I recommend using the impersonal form.
We appreciate the reviewer’s insightful comment. To prevent unintended attribution and to
clarify our intended message, we have revised the sentence using neutral, impersonal phrasing.
“Meanwhile, parallel health-economic modelling demonstrates how off-the-shelf constructs
shorten operating time, reduce rehab visits, and delay costly arthroplasty, while also
streamlining regulatory review and expediting reimbursement decisions by health systems,
thereby improving their clinical adoption.”
Many abbreviations, including rarely used ones, are not deciphered either in the text or in the
abbreviations section.
The Authors thank the reviewer for the observation and expanded the Abbreviations list to
include all terms used.

Round 2
Reviewer 2 Report
Comments and Suggestions for Authors
The authors have noa answered all the reviewer's comments
Reviewer 3 Report
Comments and Suggestions for Authors
Accepted
Reviewer 4 Report
Comments and Suggestions for Authors
Thanks for the correction and all the new inputs.